# Targeting Alzheimer’s Disease: The Critical Crosstalk between the Liver and Brain

**DOI:** 10.3390/nu14204298

**Published:** 2022-10-14

**Authors:** Zhihai Huang, Hung Wen (Kevin) Lin, Quanguang Zhang, Xuemei Zong

**Affiliations:** Department of Neurology, Louisiana State University Health Sciences Center, 1501 Kings Highway, Shreveport, LA 71103, USA

**Keywords:** Alzheimer’s disease, amyloid-beta, liver, low-density lipoprotein receptor-related protein-1 (LRP-1), fibroblast growth factor 21 (FGF21)

## Abstract

Alzheimer’s disease (AD), an age-related neurodegenerative disorder, is currently incurable. Imbalanced amyloid-beta (Aβ) generation and clearance are thought to play a pivotal role in the pathogenesis of AD. Historically, strategies targeting Aβ clearance have typically focused on central clearance, but with limited clinical success. Recently, the contribution of peripheral systems, particularly the liver, to Aβ clearance has sparked an increased interest. In addition, AD presents pathological features similar to those of metabolic syndrome, and the critical involvement of brain energy metabolic disturbances in this disease has been recognized. More importantly, the liver may be a key regulator in these abnormalities, far beyond our past understanding. Here, we review recent animal and clinical findings indicating that liver dysfunction represents an early event in AD pathophysiology. We further propose that compromised peripheral Aβ clearance by the liver and aberrant hepatic physiological processes may contribute to AD neurodegeneration. The role of a hepatic synthesis product, fibroblast growth factor 21 (FGF21), in the management of AD is also discussed. A deeper understanding of the communication between the liver and brain may lead to new opportunities for the early diagnosis and treatment of AD.

## 1. Introduction

Alzheimer’s disease (AD), the most common type of dementia, is a growing global health concern with enormous implications for individuals and society. There are currently 40 million people living with AD worldwide; this number is estimated to rise more than threefold by 2050 [1]. According to the latest report, the annual total cost of AD treatment is predicted to reach $2.54 trillion in 2030 and $9.12 trillion in 2050 [2]. As a progressive neurodegenerative disease with complex pathobiology, AD is characterized clinically by progressive cognitive impairment and pathologically by extracellular accumulation of amyloid-beta (Aβ) plaques, intracellular neurofibrillary tangles (NFTs) composed of highly phosphorylated tau proteins, and loss of neurons and synapses [1,3]. Unfortunately, the efficient disease-modifying therapeutic approach for AD is unavailable and there is an urgent need to advance our understanding of this disease process.

It is believed that the imbalance between Aβ production and clearance plays a pivotal role in AD pathogenesis. Therefore, strategies to scavenge Aβ from the brain are considered a priority in the treatment of AD [4,5]. Unfortunately, immunotherapy targeting the central clearance of Aβ shows limited efficacy, and several adverse effects have been reported in clinical trials, such as neuroinflammation, neuronal hyperactivity, and enhanced neurotoxicity linked to Aβ oligomerization [6,7]. This fact prompted more investigators to focus on the development of peripheral Aβ clearance-based strategies, a safer therapy [6,8]. Indeed, most of the brain Aβ can be cleared via transportation to the periphery [9,10]. Although it is still unclear how Aβ is cleared peripherally, several pathways have been proposed to mediate the efflux of brain Aβ into the periphery, including the blood–brain barrier (BBB) and perivascular lymphatic drainage pathway [11,12]. After some Aβ peptides are eliminated through phagocytosis or proteolytic degradation in the central nervous system (CNS), other Aβ peptides can be transferred into the blood through the BBB, in interstitial or cerebrospinal fluid. Some of these Aβ peptides are degraded by phagocytosis or Aβ-degrading enzymes, while others are carried by lipoproteins, erythrocytes, and albumin to peripheral organs or tissues, where they are degraded by hepatocytes or macrophages, or excreted by the kidneys and liver [12,13]. Apart from the classical amyloid hypothesis, increasing evidence suggests that AD may be a systemic metabolic syndrome [14,15,16]. Interestingly, recent studies have highlighted that the liver, a principal organ responsible for system-wide metabolic control and metabolic detoxification, contributes significantly to Aβ clearance [10,11,17], and hepatic dysfunction may increase the risk of developing AD [18,19,20]. This review aimed to discuss recent advances in our understanding of the crosstalk between the liver and AD pathology and summarize the potential of relevant therapeutic strategies for disease prevention and treatment. We proposed that liver dysfunction may contribute to AD pathology through several pathophysiological pathways (Figure 1). 

## 2. Background of Alzheimer’s Disease

AD is an age-related neurodegenerative disorder, with an insidious onset and deadly outcome. The clinical presentation of AD comprises memory loss and difficulties with thinking, speech, completing familiar tasks, and problem-solving skills [21,22]. The onset and progression of AD involves complex interactions of multiple factors. Largely, AD can be divided into early-onset familial forms and late-onset sporadic forms. Early-onset AD is strongly linked to mutations in the presenilin 1, presenilin 2, or amyloid precursor protein (APP) genes. In contrast, the large majority of AD cases are the sporadic form (>95%), which is thought to be induced by abnormal processing of Aβ [23,24]. Apart from aging, several risk factors contributing to sporadic AD have been proposed such as systemic inflammation, vascular disease, sleep disturbances, and gut microbiota dysbiosis [25,26,27,28]. The accumulation of Aβ and its oligomer in the brain is normally considered to be a key contributor to the pathogenesis of AD, but recently, several other mechanisms have also been proposed to explain its disease progression [22,29].

AD exhibits similar pathophysiological characteristics to type II diabetes mellitus, obesity, and other metabolic syndromes, including oxidative stress, abnormal fasting glucose, and insulin resistance [30,31]. Therefore, it has been suggested that AD should be considered in the context of metabolic disorders, and some even suggest terming AD as “Type 3 diabetes mellitus” [32,33,34]. Moreover, the role of several peripheral organs, including the liver and kidney, in facilitating circulating Aβ clearance and systemic metabolic regulation, has been increasingly recognized [11,35]. In this context, coupled with the limited success of strategies aimed to scavenge Aβ peptides centrally, therapeutic targets for AD are no longer solely based on past knowledge of this disease. With the emergence of novel treatment targets, treatment modalities for AD have become diversified. In the following sections, we discuss the contribution of peripheral organs in Aβ clearance, and the metabolic disruptions observed in AD.

### 2.1. Central and Peripheral Aβ Catabolism

Various theories have been proposed to explain the pathogenesis of AD, including the amyloid, tau hyperphosphorylation, and mitochondrial dysfunction hypotheses [14,36,37,38]. Among them, the amyloid hypothesis remains the most accepted model to explain AD pathogenesis [39]. According to this theory, the imbalance between Aβ production and clearance could drive the sequential cleavage of APP by β and γ secretase enzymes in the brain, resulting in the accumulation of pathological forms of Aβ that subsequently form insoluble amyloidogenic fibers and aggregate into plaques [36,40]. Moreover, this aggregation will lead to kinase activation, which will result in microtubule-associated τ protein hyperphosphorylation and aggregation into insoluble NFTs [1]. Microglia infiltration in response to these aggregates can also mediate synapse loss by engulfing synapses, thereby exacerbating AD pathology, and ultimately contributing to dementia symptoms [41,42]. Unfortunately, immunotherapies designed to centrally scavenge Aβ have failed to modify disease symptoms, possibly due to the potential adverse effects of these treatments and the limited central clearance rate of Aβ. Recent studies found that peripheral organs contribute significantly to clearing Aβ accumulation in the brain. Experiments using radiation technology have shown that more than 60% radioactivity could be detected in peripheral organs, including the liver, kidney, gastrointestinal tract, and skin, after I125-labeled Aβ was administered intravenously or intracranially [9,10,43]. Additionally, peritoneal dialysis, a therapeutic method for removing metabolites and wastes from the blood, can significantly lower the plasma Aβ levels in patients and APP/PS1 mice, attenuate AD pathology (such as Tau hyperphosphorylation, glial activation, neuroinflammation, neuronal loss, and synaptic dysfunction), and increase Aβ phagocytosis in microglia, as well as rescue cognitive impairment in the animals [44]. It has also been reported that unilateral nephrectomy increased Aβ deposition in the brain and exacerbated AD pathologies, yet treatment with furosemide (a type of loop diuretic mainly acting on the kidney) exerted protective effects, as evidenced by the reduced Aβ levels in the blood and brain, with ameliorated AD symptoms and cognitive deficits [45]. These promising results have sparked an increased interest in developing strategies for the peripheral clearance of Aβ.

### 2.2. Metabolic Dysfunction in Alzheimer’s Disease

Remarkably, mounting evidence indicates that AD patients usually develop central and peripheral metabolic dysfunction [15,46,47,48]. The human brain consumes 20% of the body’s energy consumption at rest, although it represents only 2% of body weight; neurons expend 70–80% of total brain energy [49]. As a high energy-demand organ, the brain is sensitive to fuel supply and energy changes. Moreover, these metabolic alternations can significantly affect the onset and progression of multiple neurodegenerative diseases [49,50]. Glucose is the main energy substrate for the brain, and since brain neurons cannot synthesize or store glucose directly, their energy supply primarily depends on the continuous uptake of external glucose. Cerebral glucose metabolism involves two key processes: glucose transport and intracellular oxidative catabolism. The later involves complex pathways, the tricarboxylic acid (TCA) cycle and oxidative phosphorylation, which occur within the mitochondria, and the glycolysis and pentose phosphate pathway in the cytoplasm [14,51]. Since glucose is a polar hydrophilic molecule that cannot freely cross the BBB, normal physiological glucose transport in the brain is strongly dependent on the specific transport proteins, including sodium-dependent glucose transporters (SGLTs) and sodium-independent glucose transporters (GLUTs) [52,53]. Through intracellular catabolism, glucose transported to the brain is eventually converted into adenosine triphosphate (ATP) and its related metabolites to fuel neural activity and biosynthesis.

Consistent evidence from human studies and animal studies has revealed abnormal glucose metabolism in the brain that could occur prior to AD pathology and cognitive impairment, including reduced GLUT1 and GLUT3 [54,55], impaired insulin signaling [56,57], aberrant glucose metabolism [58,59,60], mitochondrial dysfunction and reduced ATP synthesis [61,62]. Furthermore, a significant reduction in cerebral glucose metabolism can be detected years or even decades before clinical symptoms appear in presenilin-1 gene mutation carriers [63]. In rat and Drosophila AD models, both recovery of the insulin signaling pathway, which is critical in regulating glucose transport across membranes, and overexpression of GLUT1 in neurons can effectively slow the progression of AD-related pathology [64,65]. These findings emphasize the role of aberrant brain energy metabolism in AD pathogenesis. Intriguingly, besides its role in facilitating Aβ clearance, the liver is the primary organ for glucose production and storage, and a normal hepatic function also markedly contributes to the maintenance of systemic glucose metabolism [66]. In fact, most AD patients exhibit fasting hyperglycemia, which may serve as a contributor to cerebral glucose dysregulation [33,67,68]. Furthermore, systemic administration of AD vectors directly impaired insulin signaling in the liver, which diminished the ability of insulin to inhibit hepatic glucose production, thus disrupting systemic glucose homeostasis [69]. Exposure to chronic hyperglycemia also exacerbates AD neurodegeneration and cognitive impairment in AD animals [70,71]. Notably, a large body of preclinical research and prospective cohort studies indicate that liver metabolic alterations or dysfunction could precede progressive cognitive decline and other typical symptoms of AD [20,72,73]. The liver is the first organ to exhibit metabolic dysfunction with AD progression [74]. Abnormal liver function enzymes were associated with a significantly increased AD risk [75]. The liver tissue from AD patients also contained less Aβ than that of healthy individuals, suggesting an impaired ability of the liver to uptake circulating Aβ [18]. These findings provide critical evidence that liver dysfunction as an early pathological event in AD, and will be discussed thoroughly later in this review.

## 3. Hepatic Metabolism in Health and Disease

The liver is an essential hub for numerous physiological processes, primarily responsible for system-wide metabolic regulation, metabolic detoxification, protein synthesis, immune system support, and lipid and cholesterol homeostasis [76]. Moreover, the liver’s ability to produce and store glycogen as a readily available glucose reserve for the body is vital. Upon feeding, the decline in glucagon and increase in insulin, followed by the increase of glycolysis and glycogen deposition, drive the liver to shift to the net glucose intake mode [16,76]. By contrast, as the body changes from the absorptive state to the fasting state, insulin diminishes, and glucagon increases, turning the liver into promoting net glucose output to meet the energy demands of the brain and other organs [76,77]. Liver-derived glucose production could represent 90% of endogenous glucose production and is of particular importance for cells that partially or completely rely on glucose as energy substrates, such as neurons, erythrocytes, and renal medullary cells [78,79]. These liver-associated physiological processes, however, are dysregulated in many metabolic disorders and AD [66,80]. During the fed state, insulin stimulates glucose absorption into muscle and adipose tissue and inhibits hepatic gluconeogenesis. In the case of hepatic insulin resistance, impaired suppression of hepatic glucose production can lead to hyperglycemia and compromises cerebral glucose transport [66,81,82]. Therefore, well-coordinated hepatic glucose metabolism is crucial for health.

Importantly, as suggested above, the liver is the principal organ responsible for Aβ peripheral clearance in normal physiological conditions. When effluxed from the brain, Aβ binds to other molecules. For instance, circulating Aβ can be transported to the liver via high-density lipoprotein (HDL) particles, and subsequently taken up into cells by hepatocyte low-density lipoprotein receptor-related protein-1 (LRP-1), where it is degraded or cleared through bile excretion [11,83,84]. Abnormalities in these processes have been observed in both AD patients and animal models. Notably, an earlier study reported that the liver tissue from AD patients contains fewer Aβ molecules than that of healthy individuals, implying that the liver’s Aβ clearance function may be impaired in AD pathology [18]. Indeed, it is increasingly recognized that hepatic dysfunction is an early event in AD, preceding the pathological characteristics. In the following section, we discuss recent findings supporting hepatic dysfunction as a risk factor in AD.

## 4. Liver Dysfunction and AD Pathology

### 4.1. Animal Studies on the Crosstalk between Liver Dysfunction and AD Pathology

The role of liver dysfunction in aging and AD is beginning to receive attention. The contribution of liver dysfunction to the pathogenesis of AD is well-described in animal models (Table 1). LRP-1 is an endocytic and signaling receptor expressed in multiple tissues, including LRP-1 existing on the cell surface and soluble form of LRP1 (sLRP1) in plasma [59,85]. This receptor is required for the cerebral and systemic clearance of Aβ. In the brain, Aβ is combined with apolipoprotein E (ApoE) and transported to peripheral blood through LRP-1. Subsequently, most circulating Aβ binds to sLRP-1 to avoid excessive free Aβ concentrations in the blood [84,85]. In the liver, Aβ can be absorbed by hepatocytes via surface LRP-1 and then is eliminated. Nevertheless, both LRP-1 expression in the liver and hepatic Aβ uptake can be significantly reduced in aging and AD animals, suggesting impaired Aβ peripheral transport and clearance [86,87].

In light of the growing body of research suggesting that obesity and high-fat diets may increase the risk of AD, Kim et al. performed an important study to evaluate the relevance of non-alcoholic fatty liver disease (NAFLD) to AD pathology. They demonstrated that chronic NAFLD induced advanced pathological hallmarks of AD, including glial cell activation, neuroinflammation, neuronal apoptosis, and β-amyloid plaque load, in both wild-type and APP-Tg mice [88]. Intriguingly, decreased brain expression of LRP-1 was also observed in these animals. Additional evidence indicates that a Western diet feeding accelerated amyloid pathology in APPswe mice, which is paralleled by hypercholesterolemia and fatty liver [90]. Moreover, using an analytical method for in-depth tissue metabolome profiling, Wang et al. revealed remarkable differences in metabolome changes in the liver and brain in 5xFAD mice compared to wild-type mice [89]. In a similar manner, Zheng et al. investigated the metabolic changes in the liver, kidney, and heart of APP/PS1 mice with aging. They found that during the progression of AD, metabolic disorders existed in the peripheral organs of APP/PS1 mice, among which the liver was the first organ affected, mainly involving disturbances in energy metabolism, amino acid metabolism, nucleic acid metabolism, as well as ketone and fatty acid metabolism [74]. Furthermore, abnormal cholesterol levels can affect Aβ synthesis, clearance, and neurotoxicity. Elevated serum cholesterol levels have been shown to positively associate with an increased risk of dementia, while certain cholesterol-lowering therapies can reduce the production of Aβ [91,92]. Results from another study, which measured hepatic cholesterol and its metabolites in animals, revealed a fundamental defect of liver cholesterol in APP/PS1 and App^NL-G-F^ mice; in particular, impaired bile acid synthesis was observed [72]. Since bile acid synthesis plays a pivotal role in the excretion of cholesterol, these results collectively suggest that the overall hepatic metabolic profile is compromised in the progression of AD.

### 4.2. Clinical Data on the Crosstalk between Liver Dysfunction and AD or Dementia

Apart from animal studies, the interactions between liver dysfunction and AD/dementia have been extensively investigated in some human studies (Table 2). An early autopsy study observed that Aβ levels (Aβ40 and Aβ42) in the white and gray matter from AD patients are significantly higher than that from non-dementia subjects, and the liver tissue from AD patients contains less Aβ, suggesting Aβ clearance, particularly Aβ clearance by the liver, might be compromised [18]. Notably, this study also measured plasma Aβ42/40 ratios in AD patients, but they varied widely among individuals. This distinction could be due to age and/or gender differences of the subjects as well as other potential variables. Intriguingly, cirrhosis patients were found to have higher plasma A40 and A42 levels than healthy individuals, which were further elevated in cirrhosis patients with hepatitis B virus infection [20]. Furthermore, this study also examined blood biochemical parameters to assess hepatic function. Multiple linear regression results revealed a positive association between aberrant hepatic function and elevated plasma Aβ levels, suggesting liver dysfunction may lead to reduced peripheral Aβ clearance by the liver [20].

Several studies examined the association between altered liver enzymes and AD dementia diagnosis. In a recent study, serum hepatic function markers, cognitive performance, and AD pathological profiles were measured in 1581 AD Neuroimaging Initiative participants, with an 8 year follow-up. The results of the regression analysis revealed a strong association between elevated aspartate aminotransferase (AST) to alanine aminotransferase (ALT) ratio (peripheral liver function markers), lower levels of ALT, and AD diagnosis [73]. In particular, lower levels of ALT were correlated with increased Aβ deposition and exacerbated brain atrophy, while higher AST to ALT ratios were significantly correlated with reduced glucose metabolism in some brain regions, including the bilateral frontal, parietal, and temporal lobes. Similarly, in a longitudinal study designed to establish the association of ALT and AST with the risk of dementia, low ALT and AST levels were found to be related to a higher risk of dementia in an average 18.3 year follow-up [75]. Notably, a population-based cross-sectional survey conducted by Chen et al. [93] pointed to cirrhosis as one of the co-morbidities significantly associated with mild cognitive impairment and dementia (OR 3.29, 95% CI 1.29–8.41). Given the progressive increase in liver fibrosis with aging, Solfrizzi et al. evaluated the link between NAFLD fibrosis score and risk of dementia. Their preliminary results suggest that older adults with high NAFLD fibrosis scores do not show a markedly increased risk of dementia [94]. Nevertheless, at the subsequent 8 year follow-up, frail older adults with high fibrosis scores were at an increased overall risk of developing dementia.

Taken together, although such studies remain quite limited, these findings support liver dysfunction as an early event related to AD and might be implicated in the pathogenesis of AD. Further prospective studies with a larger sample size are needed to investigate the correlation between liver dysfunction/disease and the risk of AD.

## 5. The Liver as a Potential Therapeutic Target for Alzheimer’s Disease

### 5.1. Strategies to Enhance Hepatic Aβ Degradation

A better understanding of the link between liver dysfunction in the pathogenesis of AD may present new opportunities for its early diagnosis and management. For this purpose, many attempts have been made to explore the therapeutic potential of improving liver function in the treatment of AD treatment. LRP-1, the main cell surface receptor involved in cerebral and systemic toxic Aβ clearance, is thought to be one of the therapeutic targets for AD [84]. Several measures aimed to restore or enhance LRP1-driven Aβ regulatory systems have shown great potential in mitigating AD pathological processes. For instance, fluvastatin treatment increased LRP-1 expression levels at the BBB, resulting in increased Aβ clearance and diminished accumulation in a transgenic mouse model of AD [95]. Low-level administration of recombinant LRP cluster IV has been proven to facilitate Aβ clearance, and alleviate pathological alternations and functional disorders, but this therapy did not affect phosphorylated LRP-1 levels in the brain and liver [96]. Importantly, portal insulin injection facilitated LRP-1 transport to the hepatic plasma membrane and enhanced LRP-1-dependent Aβ uptake by the liver [97]. Intriguingly, orally administered extracts of *Withania somnifera* (WS), a drug plant widely used for the treatment of neurological disorders, have been reported to reverse Aβ plaque burden and behavioral deficits in middle-aged transgenic AD mice [87]. The positive effects of WS largely depended on the modulation of hepatic LRP. Although upregulation of plasma slurp and hepatic LRP was observed after WS administration, the extract did not induce changes in brain LRP. Significantly, selective downregulation of hepatic LRP eliminated these therapeutic benefits [87]. These lines of evidence appear to corroborate the efficacy of enhanced hepatic Aβ clearance in slowing AD progression.

Moreover, regular exercise is an effective intervention in the management of diabetes mellitus and AD. Previous studies by our group and others have reported the positive effects of exercise interventions in preventing the progression of AD [98,99,100]; exercise also shows the potential to improve liver metabolic function [101,102]. To be noted, in an experimental model of AD, treadmill exercise reversed diminished plasma sLRP-1, as well as compromised LRP-1 mRNA and protein content in the liver, aside from a reduction in brain Aβ plaque load and up-regulation of hippocampal LRP-1 [103]. In this regard, exercise could be a powerful means to facilitate both central and peripheral Aβ clearance, with it also being the most cost-effective.

### 5.2. Strategies to Regulate Hepatic Synthesis Products

Some liver-derived synthesis products have also been reported to alleviate AD disease progression [104,105]. Fibroblast growth factor 21 (FGF21) is a peptide hormone highly synthesized in the liver and functions to maintain energy homeostasis in multiple tissues [106,107]. The physiology of FGF21 is complicated, as it can be synthesized in multiple organs and act on multiple target tissues, including the liver, brain, heart, and adipose tissue [107]. Under most physiological conditions, circulating FGF21 is predominantly derived from the liver. The specific ablation of hepatocytes using albumin Cre leads to a dramatic decrease in circulating levels of FGF21 [108]. Furthermore, systemically administered FGF21 can cross the BBB in a non-saturable manner and maintain integrity [109]. Since FGF21 receptors (FGFRs), including FGFR1, FGFR2, and FGFR3, as well as an obligate coreceptor, β-klotho, are expressed in the brain, it has been suggested that circulating FGF21 may exert many positive effects on the brain [107]. Additionally steatosis, inflammation, hepatocyte damage, and fibrosis in the liver can develop from FGF21 deficiency [110,111,112], whereas FGF21 analogue administration or upregulation of its endogenous level attenuates these processes [113,114]. FGF21 has attracted growing interest as a therapeutic agent for numerous metabolic syndromes including obesity and diabetes [106,115]. More importantly, compared with healthy subjects, the plasma FGF21 levels were significantly lower in AD patients, suggesting impaired hepatic ability to synthesize FGF21 in AD pathology [116].

Given the large body of research suggesting a strong link between AD/dementia and metabolic disease, a seminal study by Chen et al. investigated the effects of FGF21 on AD pathology [104]. After an intracranial injection of Aβ_25-35_, the rats were injected subcutaneously with FGF21 (twice daily for 8 days). The results showed that FGF21 exerted numerous protective effects, including alleviation in Aβ_25-35_-induced neuronal apoptosis, tau pathology, mitochondrial damage, and oxidative stress, paralleled by a significant improvement in the animals’ cognitive deficits [104]. Furthermore, previous studies demonstrated that the astrocyte–neuronal–lactate shuttle (ANLS) is critical for memory formation and consolidation [117,118]. Abnormalities in this communication have been identified in AD, including reduced lactate content, down-regulation of GLUTs and monocarboxylate transport proteins (MCTs), and lower lactate dehydrogenase (LDHA) in the cortex and hippocampus [119,120,121]. Based on these findings, the latest study from the same group revealed that FGF21 ameliorates neurodegeneration in transgenic AD animals through modulation of the ANLS system [105]. Consistent with previous reports, in both in vivo and in vitro models of AD, a compromised ANLS was found, and FGF21 delivered centrally or peripherally alleviated AD-related degeneration in mice; blocking FGFR1, however, prevented the beneficial effects induced by FGF21, including the alleviation of tau pathology and the upregulation of ATP, MCT2, and MCT4 in the brain [105]. Interestingly, ablation of MCT2 and MCT4 also eliminated the beneficial effect of FGF21 on cortical and hippocampal ATP levels, suggesting that ANLS regulation could be a target for the neuroprotection of FGF21. Intriguingly, in a rodent model of neurodegeneration, caloric restriction, a commonly adopted dietary intervention in preclinical research, significantly increased the mRNA expression of FGF21 in the liver [122]. This is in line with elevated circulating FGF21 and its expression in the brain, as well as alleviated synaptic loss and cognitive deficits. Similarly, FGF21 treatment improved metabolic dysfunction-related cognitive impairment and AD-like degeneration in obese rats [123,124]. These findings emphasized the potential contribution of FGF21, a hormone primarily derived from the liver, in preventing AD-like neurodegeneration.

## 6. Limitations and Future Perspectives

More recently, the role of liver dysfunction in AD pathology has emerged as an exciting field of investigation. Encouraging findings from animal studies also reflect the translational value of developing strategies to restore hepatic physiological function for AD management. Nevertheless, given the limited literature available, many questions remain to be addressed in this nascent field. Current evidence points to a reduction in hepatic LRP-1 under AD pathology, which results in a compromised peripheral clearance of Aβ, but whether liver dysfunction directly exacerbates brain Aβ load and accompanying neurodegeneration is not yet well-defined. As one of the peripheral organs responsible for Aβ clearance, the resection of the kidney has been proven to increase Aβ burden in the brain and exacerbated associated pathologies [45]. Along these lines, as the liver is in charge of eliminating the majority of the circulating Aβ, inhibition of its ability to uptake and clear circulating Aβ, or hepatectomy removal, may shed light on these issues. In addition, the confirmation that peripheral FGF21 delivery protects against AD via cerebral FGF1 and MCTs emphasizes the essential role of circulating FGF21 for cerebral energy homeostasis [105]. Given that liver accounts for more than 80% of the FGF21 product in circulation, a novel liver-specific knockout of FGF21 mouse model has been established to identify the contribution of liver-derived FGF21 in metabolic regulation [125,126]. This transgenic model might serve as a powerful tool for understanding the potential role of liver FGF21 deficiency in AD-like pathologies. Intriguingly, FGF21 has been shown to be an exercise-responsive factor. Circulating FGF21 elevates after different forms of exercise, and the synthesis of FGF21 in response to exercise may mediate the multiple health benefits conferred by exercise [127,128]. A recent study revealed that the protective effect of exercise on cardiac function was eliminated in liver-specific FGF21 knockout mice [129]. Similar to exercise, FGF21 is protective against the development of AD; whether the induction of liver FGF21 is required for the beneficial effects of exercise for AD would be an issue worthy of further investigation.

Remarkably, although studies have revealed a connection between hepatic pathological alternations and AD, little is known about how different forms and degrees of liver pathologies are involved in AD progression. Further studies are required to elucidate these critical issues. In this sense, targeted metabolomics analyses on liver tissue from experimental animal models and the wider employment of liver disease-related animal models would be useful to identify potential pathogenic factors, as well as to determine potential differences among these pathological processes, and earlier AD markers. Since AD normally triggers irreversible tissue damage after onset, a better understanding of these physiological processes will undoubtedly facilitate the early diagnosis and management of AD.

Moreover, developing treatments targeted at liver protection will not only help us understand the peripheral pathological mechanisms of AD, but will also guide our efforts to develop innovative treatment modalities for this disease. To be noted, some liver-targeted mitochondrial uncoupling agents, such as derivatives of 2,4-dinitrophenol-methyl ether, have shown the potential to attenuate hepatic gluconeogenesis, insulin resistance, hepatic lipid peroxidation, protein carbonylation, and systemic inflammation [130,131]. In addition, J147, a novel AD candidate used to alleviate neurological dysfunction, has been reported to reduce free fatty acid levels in the liver via modulation of AMPK/ACC1 signaling [132]. Given that accumulated free fatty acid in the liver is the major driver of NAFLD and the potential link between NAFLD and AD pathology, the liver is probably an attractive target for the neuroprotection of J147. In this regard, the treatment efficacy and molecular mechanisms of these agents in AD pathobiology will deserve further investigation.

In light of the limited effect of monotherapy, combined therapies for AD are increasingly being advocated to enhance the treatment efficacy of traditional therapy. Whether the combination of traditional treatment strategies with interventions to enhance peripheral Aβ clearance or supplementation with FGF21 will have additive effects remains to be elucidated. If successfully applied in clinical practice, these combined approaches will greatly benefit patients with AD and reduce associated social and economic burdens.

## 7. Conclusions

Peripheral organs, especially the liver, are of importance for Aβ clearance, and disturbances in peripheral Aβ metabolism may contribute to AD progression. Furthermore, aberrant liver function might also undermine the communication between the liver and brain, including normal glucose metabolism, and the supply of certain key hepatic synthesis products, resulting in metabolic disorders. Improvements in our understanding of the interaction between hepatic disorders and the brain will help identify the peripheral mechanisms of AD pathogenesis and contribute to its early diagnosis and treatment. In addition, the failure of monotherapies has prompted increasing agreement that combination therapies directed to multiple targets may accumulate their beneficial effects in AD treatment. Thus, combinatorial approaches targeting the liver with other pathologies may potentially represent new AD therapeutic strategies.

## Figures and Tables

**Figure 1 nutrients-14-04298-f001:**
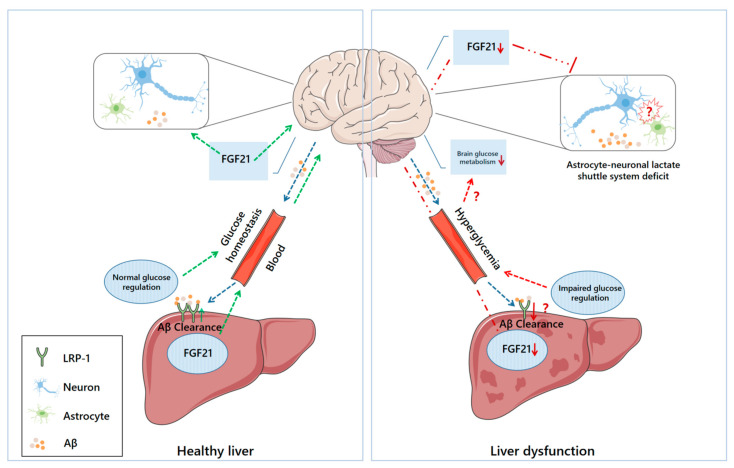
**The potential communication between liver dysfunction and AD neurodegeneration.** Under normal physiological conditions (**left**), hepatocytes uptake circulating Aβ via LRP-1 and promote Aβ clearance. FGF21 can also be processed by the liver, and then delivered to the brain via the blood, where it can exert protective effects. Circulating FGF21 may contribute to the stability of the astrocyte–neuron–lactate shuttle system and cerebral energy homeostasis. Under pathological conditions (**right**), diminished LRP-1 hinders hepatic Aβ clearance, and the liver’s ability to synthesize FGF21 may also be compromised, leading to reduced supplementation of these substances to the brain. Although it is not yet clear what drives the impaired astrocyte–neuron–lactate shuttle system in AD and whether there is a causal relationship between that and liver dysfunction, lower FGF21 may weaken its protective effect on this communication system. Additionally, normal glucose regulation in the liver is essential for maintaining systemic glucose homeostasis. Conversely, compromised hepatic glucose regulation may induce hyperglycemia and disrupt glucose metabolism in the brain.

**Table 1 nutrients-14-04298-t001:** Animal studies on the crosstalk between liver dysfunction and AD pathology.

Animal Model	Species	Age/Gender	Main Findings	Reference
-	Sprague-Dawley rats, C57BL/6J mice, and ICR mice	7 weeks and 13 months/unknown	Compared with 7-week-old rats, the expression of LRP-1 in the liver and the hepatic Aβ (1–40) uptake decreased significantly in 13-month-old rats.	[86]
AD	APP/PS1 mice, APPSwInd J20 mice, and wild-type mice	3–4 months, 9–10 months, and 23–24 months/male and female	The liver LRP and Aβ-degrading protease neprilysin, as well as plasma sLRP expression, were significantly reduced in AD mice compared to controls.	[87]
NAFLD	APP-Tg mice wild-type mice	4, 7, and 14 months/unknown	Chronic NAFLD induced advanced AD pathology in both WT and APP-Tg mice, including activated microglial cells, increased inflammatory cytokines, neuronal apoptosis, and β-amyloid plaque load. In addition, a decrease in LRP-1 brain expression was observed.	[88]
AD	5xFAD mice and wild-type mice	5 months/unknown	The amine/phenol submetabolome changes in the liver and brain tissues of 5xFAD mice were significantly different from that of wild-type mice.	[89]
AD	APP/PS1 mice and wild-type mice	1, 5, and 10 months/male	During amyloid pathology progression, the liver is the earliest organ showing metabolic dysfunction, including disturbances in energy metabolism, amino acid metabolism, nucleic acid metabolism, as well as ketone and fatty acid metabolism.	[74]
AD	APPswe mice	10 weeks/male	Western diet feeding accelerated amyloid pathology and induced advanced hypercholesterolemia, and fatty liver disease.	[90]
AD	App^NL-G-F^ mice, APP/PS1 mice, and wild-type mice	6-month/male and female	AD mice show fundamental deficiencies in hepatic cholesterol metabolism and bile acid synthesis compared to age-matched wild-type mice.	[72]

Abbreviations: AD, Alzheimer’s disease; LRP-1, Low-density lipoprotein receptor-related protein-1; NAFLD, Non-alcoholic fatty liver disease.

**Table 2 nutrients-14-04298-t002:** Clinical data on the crosstalk between liver dysfunction and AD or dementia.

Disease	Type of Study	Subjects	Main Findings	Reference
AD	Longitudinal study, post-mortem samples and living subjects	17 AD patients: 7 men and 10 women, average age, 81.4 years; 21 ND subjects: 7 men and 14 women, average age, 75.8 years; Post-mortem specimens: AD (5–10), ND (5–13)	The levels of Aβ (Aβ40 and Aβ42) in white and gray matter were significantly higher in AD patients than in ND subjects. In addition, the liver tissue from AD patients contains less Aβ than that of healthy individuals.	[18]
AD	Cohort study, post-mortem samples and living subjects	14 AD patients: 8 men and 6 women, average age, 84.6 ± 6.7 years; 9 ND subjects: 7 men and 2 women, average age, 83.9 ± 5.4 years; Post-mortem specimens: AD (n = 37), ND (n = 17)	Compared with healthy subjects, the levels of docosahexaenoic acid were reduced in the liver and multiple brain regions in AD patients. In addition, mini-mental state examination scores were positively correlated with docosahexaenoic/α-linolenic ratios in the brain and liver.	[19]
Cirrhosis	Cohort study, living subjects	46 cirrhosis patients and 46 age and gender-matched healthy subjects.	Plasma Aβ levels were positively correlated with impaired hepatic function, including elevated bilirubin, globulin, AST, and AST/ALT ratio, and decreased albumin and A/G ratio.	[20]
MCI and dementia	Cross-sectional study, living subjects	1576 MCI and 697 dementia patients, >65 years	Cirrhosis is one of the most correlated comorbidities with MCI and dementia.	[93]
AD and MCI	Cohort study, living subjects	1581 participants (included 407 cognitively normal older adults), 884 men and 697 women, average age, 73.4 ± 7.2 years	AD diagnosis was significantly associated with elevated AST/ALT ratio and decreased levels of ALT. In addition, lower levels of ALT were associated with increased amyloid-β deposition, decreased cerebral glucose metabolism, and greater atrophy.	[73]
Liver fibrosis	Longitudinal study, living subjects	1061 older adults, 65 to 84 years old	Older adults with only a high NAFLD fibrosis score did not have a significantly increased risk for dementia. However, during the eight years of follow-up, frail older adults with a high fibrosis score showed an increased overall risk of dementia.	[94]
Dementia	Longitudinal study, living subjects	1857 dementia patients	Lower ALT levels (<10%) were strongly associated with an increased risk of dementia. In addition, a similar pattern was observed with lower AST levels, but with a smaller magnitude.	[75]

Abbreviations: AD, Alzheimer’s disease; AST, Aspartate aminotransferase; ALT, Alanine aminotransferase; MCI, Mild cognitive impairment; ND, Non-demented; NAFLD, Non-alcoholic fatty liver disease.

## Data Availability

Not applicable.

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
