# Peer review of "Targeting Alzheimer’s Disease: The Critical Crosstalk between the Liver and Brain"

_nutrients, 2022, doi:10.3390/nu14204298_

Round 1
Reviewer 1 Report
The manuscript reviewed recent animal and clinical findings indicating that liver dysfunction represents an early event in AD pathophysiology. Tt is helpful for early diagnosis and treatment of AD. The manuscript can be published in the journal with minor revised.
1. The liver metabolic dysfunction of Alzheimer's Disease should be discussed in the part of 2.2.Metabolic dysfunction in Alzheimer's Disease
Author Response
Review 1
The manuscript reviewed recent animal and clinical findings indicating that liver dysfunction represents an early event in AD pathophysiology. It is helpful for early diagnosis and treatment of AD. The manuscript can be published in the journal with minor revised.
Authors' response: We appreciate the thoughtful assessment from the reviewer, and all the suggestions have been carefully addressed.
1.The liver metabolic dysfunction of Alzheimer's Disease should be discussed in the part of 2.2.Metabolic dysfunction in Alzheimer's Disease
Authors' response: We appreciate this excellent suggestion. As suggested by the reviewer, the content about the liver metabolic dysfunction of Alzheimer's Disease was included in this revision (Line 159-167).

Reviewer 2 Report
This review discusses recent advances in our understanding of the crosstalk between the liver and Alzheimer’s disease pathology and summarizes relevant therapeutic strategies for disease prevention and treatments. The topic is interesting and has a merit to draw much attention from wide range of readers of this field. This reviewer has no major objection against this manuscript’s point of view, although a few issues should be pointed out.
There are several lines of evidence which show positive effects of exercise on liver function and Alzheimer’s disease. Regular exercise has been well documented as a very effective measure to prevent not only metabolic disorders including diabetes mellitus but Alzheimer’s disease. Although direct evidence for a causal relationship between liver dysfunction and Alzheimer’s disease, it may be worthwhile to touch upon these issues in this review, thus strengthening somewhat the point of this review.
Minor issues
Line 73, Ab should read as Ab?
Author Response
Review 2
This review discusses recent advances in our understanding of the crosstalk between the liver and Alzheimer's disease pathology and summarizes relevant therapeutic strategies for disease prevention and treatments. The topic is interesting and has a merit to draw much attention from wide range of readers of this field. This reviewer has no major objection against this manuscript's point of view, although a few issues should be pointed out.
There are several lines of evidence which show positive effects of exercise on liver function and Alzheimer' s disease. Regular exercise has been well documented as a very effective measure to prevent not only metabolic disorders including diabetes mellitus but Alzheimer's disease. Although direct evidence for a causal relationship between liver dysfunction and Alzheimer's disease, it may be worthwhile to touch upon these issues in this review, thus strengthening somewhat the point of this review.
Authors' response: Thanks for the holistic review and the suggestions for improving the quality of the manuscript. We have now addressed these issues in the manuscript.
- We agreed with thereviewer's viewpoint that exercise could be a powerful intervention to improve liver function and against Alzheimer' s disease. These contents were included and discussed in this revision (Line 310-318; 388-395).
- In this revision, more extensive discussionsof these points were added to the Discussion section (Line 396-405).
Minor issues
Line 73, Ab should read as Aβ?
Authors' response: This statement is now rewritten in this revision (Line 74).
